# New Anti-inflammatory Flavonol Glycosides from *Lindera akoensis* Hayata

**DOI:** 10.3390/molecules24030563

**Published:** 2019-02-04

**Authors:** Chung-Ping Yang, Pei-Hsin Shie, Guan-Jhong Huang, Shih-Chang Chien, Yueh-Hsiung Kuo

**Affiliations:** 1Key Laboratory of Preventive Veterinary Medicine and Biotechnology, Longyan University, Longyan 364012, China; cpyang218@gmail.com (C.-P.Y.); sps0220@gmail.com (P.-H.S.); 2Fujian Provincial Key Laboratory for the Prevention and Control of Animal Infectious Diseases and Biotechnology, Longyan University, Longyan 364012, China; 3Department of Chinese Pharmaceutical Sciences and Chinese Medicine Resources, China Medical University, Taichung 404, Taiwan; gjhuang@mail.cmu.edu.tw; 4The Experimental Forest Management Office, National Chung-Hsing University, Taichung 402, Taiwan; 5Department of Biotechnology, Asia University, Taichung 413, Taiwan; 6Chinese Medicine Research Center, China Medical University, Taichung 404, Taiwan; 7Research Center for Chinese Herbal Medicine, China Medical University, Taichung 404, Taiwan

**Keywords:** *Lindera akoensis*, flavonol glycoside, anti-inflammatory

## Abstract

Inflammation is related to many diseases. *Lindera akoensis* Hayata was often used in folk therapy in Taiwan for inflammation. In this study, three new flavonol acyl glycosides, namely kaempferol-3-*O*-β-*D*-4′′,6′′-di-(*E*)-*p*-coumaroylglucoside (**1**), 3′′-(*E*)-*p*-coumaroylafzelin (**2**) and 4′-*O*- methyl-2′′,4′′-di-(*E*)-*p*-coumaroylquercitrin (**3**), and three components, 3β-dodecyl-4β-hydroxy- 5β-methyldihydrofuran-2-one (**4**), 2β-acetoxyclovan-9α-ol (**5**), (1α,4β,6β)-trihydroxyeudesmane (**6**) that were isolated from the natural product for the first time were obtained along with 25 known compounds from *L. akoensis*. Their structures were determined by comprehensive spectroscopic analyses (1D and 2D NMR, EI-, ESI- and HRESI-MS). The ability of **1** to decrease the LPS-stimulated production of nitrite in RAW264.7 cell was evaluated, showing an IC_50_ value of 36.3 ± 3.2 μM. This result supports the value of *L. akoensis* as a traditional medicine resource.

## 1. Introduction

*Lindera akoensis* Hayata belonging to the Lauraceae family, moreover, it is an endemic species widely distributed in central and southern Taiwan. Traditionally it has been used by local residents to treat various inflammation symptom [1]. The genus *Lindera* has shown many bioactivities, including antitumor [2,3,4], anti-inflammatory [5,6] and antibacterial [7,8] properties in previous literature reports. Previous phytochemical research of the genus *Lindera* revealed an abundance of butalactones [8,9,10], sesquiterpenes [11] and flavonoids [9,12,13] in this genus. In our earlier study on the aerial parts of *L. akoensis*, was explored the isolation of butanolides, flavonols, and lignans [9]. In order to confirm the traditional folk usage of *L. akoensis*, this study continued our previous work on the isolation and purification of *L. akoensis* components. Six novel compounds (Figure 1) and 25 known compounds were isolated and identified. Next their anti-inflammatory activity was evaluated. Based on our experimental results, compound **1** has anti-inflammatory activity by decreasing nitric oxide (NO) production induced by lipopolysaccharide in mouse macrophage RAW264.7 cells in IC_50_ 36.3 ± 3.2 μM, having potential as a lead compound to treat symptoms of inflammation.

## 2. Results and Discussion

Thirty one compounds were isolated and identified from aerial part of *L. akoensis* including kaempferol-3-*O*-β-*D*-4′′,6′′-di-(*E*)-*p*-coumaroylglucoside (**1**), 3′′-(*E*)-*p*-coumaroylafzelin (**2**) and 4’-*O*-methyl-2′′,4′′-di-(*E*)-*p*-coumaroylquercitrin (**3**), and three compounds that were isolated from the natural product for the first time, 3β-dodecyl-4β-hydroxy-5β-methyldihydrofuran-2-one (**4**), 2β-acetoxyclovan-9α-ol (**5**), and (1α,4β,6β)-trihydroxyeudesmane (**6**) (Figure 1) along with 25 known compounds, including two monoterpenoids, 17 sesquiterpenoids, and six steroids. Their structures were elucidated by ESI-MS, UV, IR, 1D and 2D NMR spectrometry and comparisons with data from the literature.

Kaempferol-3-*O*-β-*D*-4′′,6′′-di-(*E*)-*p*-coumaroylglucoside (**1**) was isolated as a pale yellow solid. The IR spectrum showed the presence of hydroxyl (3240 cm^−1^) and carbonyl groups (1651 cm^−1^). Four structural units were observed in the 2D-NMR spectrum: two (*E*)-*p*-coumaroyls, glucose and a kaempferol nucleus (Figure 2). Two A_2_X_2_ coupling systems (δ_H_ 7.74 (2H, *d*, *J*= 8.5 Hz, H-5′′′, -9′′′), 6.81 (2H, *d*, *J*= 8.5 Hz, H-6′′′, -8′′′) and 7.26 (2H, *d*, *J*= 8.5 Hz, H-5′′′′, -9′′′′), 6.80 (2H, *d*, *J*= 8.5 Hz, H-6′′′′, -8′′′′)] and two olefinic protons δ_H_ 6.41, 7.68 (each 1H, *d*, *J*= 15.9 Hz, H-2′′′, -3′′′) and δ_H_ 6.06, 7.39 (each 1H, *d*, *J*= 15.9 Hz, H-2′′′′, -3′′′′) in the ^1^H-NMR spectrum indicated the occurrence of two (*E*)-*p*-coumaroyl group. A down-shifted chemical shift δ_H_ 5.18 (1H, *t*, *J*= 9.4 Hz) appeared on C-4′′ that conjecturing esterified on H-4′′ position, the other *(E)*-*p*-coumaroyl group replaced on CH_2_OH of glucose corroborated by HMBC spectrum. The characteristic kaempferol signals observed in the ^1^H-NMR are consistent with the literature [14], the only difference being the fact that the H-3 proton was not detected and the ^13^C-NMR signal of C-3 (δ_C_ 135.4) was more down-shifted than the C-3 of apigenin (δ_C_ 103.2) furthermore, a significant HMBC relationship of H-1′′ to C-3 was detected, so through the above evidence, the glucose-kaempferol linkage was presumed to be in a C-3-O-C-1′′ configuration. The axial-axial coupling constant (H-1′′ and -2′′, *J*= 7.8 Hz) and DEPT-135 signal of secondary carbon (C-6′′, δ_C_ 64.4) were observed, supporting the existence of β-glucose. Accordingly, the structure of **1** was elucidated as kaempferol-3-*O*-β-*D*-4′′,6′′-di-(*E*)-*p*-coumaroylglucoside, and named linderakoside F as shown in Figure 1.

3′′-(*E*)-*p*-Coumaroylafzelin (**2**) was isolated as a pale yellow solid. The IR spectrum showed the presence of hydroxyl (3431 cm^−1^) and carbonyl groups (1655 cm^−1^). Three constituents of the structure were observed—(*E*)-*p*-coumaroyl, rhamnose and kaempferol—through 2D-NMR spectrum (Figure 2). A typical methyl ^1^H-NMR signal of rhamnose δ_H_ 0.93 (3H, *d*, *J*= 5.5 Hz), different from 4′-*O*-methyl-4′′-(*E*)-*p*-coumaroylafzelin and 4′′-(*Z*)-*p*-coumaroylafzelin in our previous research [9]. The steric hindrance effect between the esterification on C-3′′ and ketone of flavone, the ^1^H-NMR signal of the methyl group (H-6′′) was down-shifted at δ_H_ 0.97 (3H, *d*, *J*= 5.5 Hz), deshielded by the ketone and the aromatic flavone ring, that demonstrated the *p*-coumaroyl position was different from 4′-*O*-methyl-4′′-*(E)*-*p*-coumaroylafzelin and 4′′-(*Z*)-*p*-coumaroylafzelin. An A_2_X_2_ coupling system at δ_H_ 7.48 (2H, *d*, *J*= 8.7 Hz, H-5′′′, -9′′′) and δ_H_ 6.81 (2H, *d*, *J*= 8.7 Hz, H-6′′′, -8′′′), as well as an olefinic proton signals at δ_H_ 6.43 and 7.72 (each 1H, *d*, *J*= 15.7 Hz, H-2′′′, -3′′′) could be observed suggesting the presence of an (*E*)-*p*-coumaroyl moiety. The α-rhamnose moiety was confirmed by the small axial- equatorial coupling constant (H-1′′ and -2′′, *brs*). Comparing with our previous studies [12], furthermore the HMBC relationship of H-1′′ to C-3 existed, so the the rhamnose-kaempferol linkage was in a C-3-O-C-1′′ configuration that was confirmed by ^13^C-NMR. Based on the above deduction, **2** was designated to be a new compound, 3′′-(*E*)-*p*-coumaroylafzelin, as shown in Figure 1, and named linderakoside G.

4′-*O*-Methyl-2′′,4′′-di-(*E*)-*p*-coumaroylquercitrin (**3**) was a pale yellow solid. The IR spectrum showed the presence of hydroxyl (3421 cm^−1^) and carbonyl group (1649 cm^−1^). Four parts of the structure were observed, two (*E*)-*p*-coumaroyls, rhamnose and tamarixetin through the 2D-NMR spectrum (Figure 2.). A typical methyl ^1^H-NMR signal of rhamnose appeared at δ_H_ 0.85 (3H, *d*, *J*= 6.2 Hz). According to the literature [15], the methyl signal should be at δ_H_ 0.95 if the –OH on the rhamnose was not esterified, and the methyl signal was high-shifted in position 4′′ confirmining the esterfication. Two A_2_X_2_ coupling system (δ_H_ 7.50 (2H, *d*, *J*= 8.6 Hz, H-5′′′, -9′′′), 6.82 (2H, *d*, *J*= 8.6 Hz, H-6′′′, -8′′′) and 7.50 (2H, *d*, *J*= 8.6 Hz, H-5′′′′, -9′′′′), 6.85 (2H, *d*, *J*= 8.6, H-6′′′′, -8′′′′)] and two olefinic protons δ_H_ 6.27, 7.55 (each 1H, *d*, *J*= 16.0 Hz, H-2′′′, -3′′′) and δ_H_ 6.42, 7.68 (each 1H, *d*, *J*= 16.0 Hz, H-2′′′′, -3′′′′) in the ^1^H-NMR spectrum indicated the occurrence of two (*E*)-*p*-coumaroyl group. This structure was similar to linderakoside E identified in our previous work [12], with a relatively downfield chemical shift on H-2′′ δ_H_ 5.55 (1H, *dd*, *J*= 3.4, 1.7 Hz) and H-4′′ δ_H_ 4.97 (1H, *t*, *J*= 9.8 Hz), that indicated esterification on these positions. Similar to **2**, the rhamnose-kaempferol linkage was in a C-3-O-C-1′′ configuration that was confirmed by ^13^C-NMR. A methoxy signal δ_H_ 3.87 (3H, s, OMe) was observed, with a significant NOESY correlation with H-5′ δ_H_ 7.15 (1H, *d*, *J*= 8.9 Hz) and obviously HMBC correlation with C-4′ δ_C_ 152.0, thence the position was determined. The α-rhamnose moiety was confirmed by small axial-equatorial coupling constant (H-1′′ and -2′′, *J*= 1.6 Hz). Based on the above deduction, **3** was designated to be the new compound 4′-*O*-methyl-2′′,4′′-di-(*E*)-*p*-coumaroylquercitrin, and named linderakoside H, as shown in Figure 1.

3β-Dodecyl-4β-hydroxy-5β-methyldihydrofuran-2-one (**4**) was isolated as a colorless solid ([α]^2^°_D_ ± 0° (c = 0.8, CHCl_3_)). Three ^1^H-NMR signals (H-3, -4, -5 δ_H_ 2.55 (1H, *m*), 4.30 (*dd*, *J* = 4.6, 3.0 Hz), 4.44 (*qd*, *J* = 6.4, 3.0 Hz)) were similar to those of 3β-((*E*)-dodec-1-enyl)-4β-hydroxy-5β-methyl- dihydrofuran-2-one and 3α-((*E*)-dodec-1-enyl)-4β-hydroxy-5β-methyldihydrofuran-2-one in our previous work [9,10]. Compound **4** has a *cis*-relationship between H-4 and -5 according to the literature comparison [16]. Eleven CH_2_-group signals were observed, where δ_H_ 1.23 (22H, *m*), one of two methyl signals δ_H_ 1.41 (3H, *d*, *J*= 6.4 Hz) was down-shifted because the influences of the –OH and lactone moiety, another methyl δ_H_ 0.87 (3H, *t*, *J*= 7.2 Hz) was typical of a -CH_2_ chain-end (Appendix A). This compound was not described in natural product before, although Lee et al obtained it by hydrogenating 3-epilitsenolide D2 in 2001 [16].

2β-Acetoxyclovan-9α-ol (**5**) was isolated as a colorless oil, with the molecular formula C_17_H_28_O_3_ from the HR-EI-MS (*m*/*z* 280.2027 [M]^+^, calcd 280.2024). The IR spectrum showed the presence of hydroxyl (3450 cm^−1^) and carbonyl groups (1738 cm^−1^). Three singlet methyl signals in the ^1^H-NMR (δ_H_ 0.89, 1.03, 0.93, each 3H, *s,* H-13, -14, -15) were characteristic of a clovane skeleton (Appendix A.). A typical acetyl group carbonyl was observed at δ_C_ 171.0 and δ_H_ 2.02 (3H, *s*). The structure of compound **5** was similar to that of clovandiol (**16**) [17] that was isolated in this work, the only difference was δ_H_ 4.83 (1H, *dd*, *J*= 8.7, 5.9 Hz) of H-4 was down shifted more than the H-4 of clovandiol (δ_H_ 3.79, 1H, *dd*, *J*= 10.5, 5.5 Hz), thence the acetyl group is speculated to be linked at this position, and the significant HMBC relationship of H-2/C-1’ confirmed this. This work is the first to describe the structure **5** in a natural product. Heymann, et al. previously obtained it by acetylating clovandiol in 1994 [18]. (1α,4β,6β)-Trihydroxyeudesmane (**6**) was isolated as colorless needle-like crystals, with the molecular formula C_15_H_28_O_3_ from HR-EI-MS (*m*/*z* 256.2029 [M]^+^, calcd 256.2010). The absolute stereo configuration of **6** was solved by X-ray single crystal diffraction (Appendix A). The IR spectrum showed the presence of a hydroxyl (3238 cm^−1^) group. The ^13^C-NMR and DEPT spectrum showed compound **6** had a eudesmane skeleton (Appendix A.). Two isopropyl methyls (δ_H_ 0.92, 1.09, each 3H, *d*, *J*= 6.6 Hz, H-13, -12), one down-shifted methyl (δ_H_ 1.34, 3H, *s*, H-15) affected by the –OH, one relatively high-shifted methyl (δ_H_ 0.94, 3H, *s*, H-14), and two relatively down-shifted signals affected by the –OH (δ_H_ 3.32, 1H, *m*, H-1, 4.33, 1H, *dd*, *J*= 11.2, 4.4 Hz, H-6) were observed in the ^1^H-NMR (Appendix A). The relative stereo configuration was decided by the NOESY spectrum, H-6/H-14/H-15 were in an axial position as defined by their significant NOESY correlations with each other. The significant correlation of H-5/H-1, -12, and -13, and the small coupling constant (4.4 Hz) between H-6 and H-7 decided the relative stereo configuration of H-1, -5, and the isopropyl. According to the literature, compound **6** was never reported as a natural product, but it was prepared by hydrolysis of pumilaside A with hesperidinase by Kitajima et al. in 2000 [19]. 

The 25 known compounds, including two monoterpenes, (*E*)-6-hydroxy-2,6-dimethylocta-2,7- dienoic acid (**7**) [20] and *trans*-sobrerol (**8**) [21], seventeen sesquiterpenes: teucladiol (**9**) [22], globulol (**10**) [23], β-dictyopterol (**11**) [24], 4β,10α-aromadendranediol-10-methyl ether (**12**) [25], 4α,10β- alloaromadendranediol-10-methyl ether (**13**) [25], 4β,10α-aromadendranediol (**14**) [26], 4α,10β- alloaromadendranediol (**15**) [25], clovandiol (**16**) [17], caryophyllenol-II (**17**) [18], humulene diepoxide A (**18**) [18], isocaryolanediol (**19**) [18], β-caryophyllene-8,9-oxide (**20**) [18], kobusone (**21**) [18], 7,8-epoxy-1(12)-caryophyllene-9α-ol (**22**) [18], 8β-hydroxy-1(12)-caryophyllene (**23**) [27], 2β-methoxyclovan-9α-ol (**24**) [28] and 8,9-dihydroxy-1(12)caryophyllene (**25**) [29] and six steroids, β-sitosterol (**26**) [30], 5-stigmasten-3β,7β-diol (**27**) [31], 5-stigmasten-3β,7α-diol (**28**) [31], 5α,8α- epidioxy-24-methylcholesta-6,9,22-trien-3β-ol (**29**) [32], 5α,8α-epidioxy-24-methylcholesta- 6,22- dien-3β-ol (**30**) [33], and 3β-hydroxystigmast-5-en-7-one (**31**) [34] were identified by comparison of their physical and reported spectroscopic data.

Caffeic acid is an effective anti-inflammatory substance. According to the literature [35,36,37,38], it inhibits inflammatory responses in many ways, including nitric oxide (NO) produced by various induction pathways, therefore the anti-inflammatory evaluation in this work used caffeic acid as positive control. linderakoside F (1) showed *in vitro* anti-inflammatory activity since it decrease the LPS-stimulated production of nitrite in RAW264.7 cell, with the IC_50_ value 36.3 ± 3.2 μM a lot better than caffeic acid (162.8 ± 5.6 μM), in addition, they have no obvious cytotoxicity at the concentration of the experiment (Figure 3). Unfortunately, the weights of linderakoside G-H (2-3) was too small to evaluate their anti-inflammatory activity.

## 3. Experimental Section

### 3.1. General Methods

The following instruments were used for obtaining physical and spectroscopic data: optical rotations P-1020 digital polarimeter (JASCO, Kyoto, Japan); IR spectra, IR Prestige-21 Fourier transform infrared specctrometer (Shimadzu, Kyoto, Japan); UV spectrum, Shimadzu Pharmaspec-1700 UV-Visible spectrophotometer; HR-ESI-MS spectra, LCQ ion-trap mass spectrometer (Finnigan, Waltham, MA , USA); melting point, MP-J3 (Yanaco, Kyoto, Japan); and ^1^H- and ^13^C-NMR spectra, DRX- 400 at 400 and 100 MHz and 500 FT-NMR spectrometer at 500 and 125 MHz, respectively (Bruker, Bremen, Germany) with TMS as an internal standard. Silica gel column chromatography was performed on silica gel (70 - 230 mesh, Merck, Darmstadt, Germany). HPLC was performed on a Shimadzu LC-6A apparatus equipped with an IOTA-2 RI-detector. A Phenomenex Luna silica (Φ 250 × 10 mm column) was used for preparative purposes (flow rate: 2.00 mL/min). Aluminum pre-coated silica gel (Merck, Kieselgel 60 F_254_) were used for TLC monitoring with visualization by spraying with a 10% solution of H_2_SO_4_ in ethanol and heating to approximately 150°C on a hotplate.

The following instruments were used for obtaining physical and spectroscopic data: optical rotations P-1020 digital polarimeter (JASCO, Kyoto, Japan); IR spectra, IR Prestige-21 Fourier transform infrared specctrometer (Shimadzu, Kyoto, Japan); UV spectrum, Shimadzu Pharmaspec-1700 UV-Visible spectrophotometer; HR-ESI-MS spectra, LCQ ion-trap mass spectrometer (Finnigan, Waltham, MA , USA); melting point, MP-J3 (Yanaco, Kyoto, Japan); and ^1^H- and ^13^C-NMR spectra, DRX- 400 at 400 and 100 MHz and 500 FT-NMR spectrometer at 500 and 125 MHz, respectively (Bruker, Bremen, Germany) with TMS as an internal standard. Silica gel column chromatography was performed on silica gel (70 - 230 mesh, Merck, Darmstadt, Germany). HPLC was performed on a Shimadzu LC-6A apparatus equipped with an IOTA-2 RI-detector. A Phenomenex Luna silica (Φ 250 × 10 mm column) was used for preparative purposes (flow rate: 2.00 mL/min). Aluminum pre-coated silica gel (Merck, Kieselgel 60 F_254_) were used for TLC monitoring with visualization by spraying with a 10% solution of H_2_SO_4_ in ethanol and heating to approximately 150°C on a hotplate.

### 3.2. Plant Material

The aerial part of *L. akoensis* was collected in Taichung, Taiwan, in July, 2008. This material was identified by Prof. Yen-Hsueh Tseng, Department of Forestry, National Chung Hsing University, Taichung, Taiwan. A voucher specimen (CMU2008-06-LA) was deposited in the School of Pharmacy, China Medical University.

### 3.3. Extraction and Isolation

The dried aerial part of *L. akoensis* (dry weight 5.9 kg) was extracted with 95% ethanol for 7 days (20 L, three times). The dried crude extract (337.8 g) was suspended in H_2_O and partitioned successively with EtOAc and *n*-BuOH. The EtOAc layer was evaporated in vacuo to yield a residue (127.8 g) that was subjected to silica gel column chromatography (particle size 70–230 mesh) and eluted with a gradient of increasing polarity with solvent of *n*-hexane/EtOAc solvent (99:1/0:100) to give 21 fractions. Fraction 16 (7.15 g) was separated using semi-preparative HPLC with the conditions (CH_2_Cl_2_/EtOAc, *v*/*v* 6:4; *n*-hexane/acetone v/v 7:3; *n*-hexane/EtOAc *v*/*v* 1:1) alternately to afford pure **1** (143.7 mg), **2** (1.1 mg), **3** (2.0 mg), and **8** (85.5 mg). Fraction 15 (1.23 g) was separated using semi-preparative HPLC with the conditions (CH_2_Cl_2_/EtOAc, *v*/*v* 6 : 4; *n*-hexane/acetone *v*/*v* 7:3; *n*-hexane/EtOAc *v*/*v* 1:1) alternately to afford pure **4** (16.9 mg), **6** (36.2 mg), **7** (71.3 mg), **15** (12.8 mg), **16** (47.7 mg), **19** (8.5 mg), **27** (46.3 mg), and **28** (55.1 mg). Fraction 11 (5.08 g) was separated using semi-preparative HPLC with the conditions (CH_2_Cl_2_/EtOAc, *v*/*v* 7:3; *n*-hexane/acetone *v*/*v* 4:1; *n*-hexane/EtOAc *v*/*v* 3:2) alternately to afford pure **5** (12.7 mg), **9** (17.2 mg), **12** (14.3 mg), **13** (16.6 mg), **14** (10.9 mg), **22** (10.8 mg), **24** (18.8 mg), and **25** (7.6 mg). Fraction 8 (10.84 g) was separated using semi-preparative HPLC with the conditions (CH_2_Cl_2_/EtOAc, *v*/*v* 4:1; *n*-hexane/acetone *v*/*v* 9:1; *n*-hexane/EtOAc *v*/*v* 7:3) alternately to afford pure **10** (9.8 mg), **11** (9.1 mg), **17** (7.2 mg), **18** (6.4 mg), **21** (9.3 mg), **23** (8.3 mg), **26** (873.5 mg), **29** (15.5 mg), and **30** (17.3 mg). Fraction 4 (0.87 g) was separated using semi-preparative HPLC with the conditions (CHCl_3_/EtOAc, *v*/*v* 8:1; *n*-hexane/acetone *v*/*v* 10:1; *n*-hexane/EtOAc *v*/*v* 4:1) alternately to afford pure **31** (62.2 mg). Fraction 3 (10.03 g) was separated using semi-preparative HPLC with the conditions (CHCl_3_/EtOAc, *v*/*v* 9:1; *n*-hexane/acetone *v*/*v* 12:1; *n*-hexane/EtOAc *v*/*v* 5:1) alternately to afford pure **20** (10.2 mg).

### 3.4. Linderakoside F *(**1**)*

Pale yellow solid; mp: 213 °C; [α]^20^_D_-54.6° (*c* = 8.9, CH_3_OH); IR (film) ν_max_: 3240, 2936, 1651, 1605, 1512, and 1173 cm^−1^; UV ν_max_ (MeOH) nm (log *ε*): 314 (4.79), 248 (4.30) and, 210 (4.70); ^1^H-NMR (500 MHz, methanol-*d*_4_): Table 1; ^13^C-NMR (125 MHz, methanol-*d*_4_): Table 2; and positive-ion HR-ESI-MS: *m/z* 763.1630 [M + Na]^+^ (calcd for C_39_H_32_O_15_Na: 763.1633).

### 3.5. Linderakoside G *(**2**)*

Pale yellow solid; mp: 162 °C; [α]^20^_D_-149.1° (*c* = 0.6, CH_3_OH); IR (film) ν_max_: 3431, 1651, 1618 and 1171 cm^−1^; UV ν_max_ (MeOH) nm (log *ε*): 314 (4.53), 276 (4.40), 267 (4.46), 246 (4.22), and 210 (4.56); ^1^H-NMR (500 MHz, methanol-*d*_4_): Table 1; ^13^C-NMR (125 MHz, methanol-*d*_4_): Table 2; and positive-ion HR-ESI-MS: *m/z* 601.1317 [M + Na]^+^ (calcd for C_30_H_26_O_12_Na: 601.1316).

### 3.6. Linderakoside H *(**3**)*

Pale yellow solid; mp: 221 °C; [α]^2^^0^_D_-166.2° (*c* = 0.3, CH_3_OH); IR (film) ν_max_: 3421, 2933, 1649, 1605, 1512, and 1169 cm^−1^; UV ν_max_ (MeOH) nm (log *ε*): 315 (4.70), 274 (4.47), 267 (4.49), 246 (4.34), and 210 (4.70); ^1^H-NMR (500 MHz, methanol-*d*_4_): Table 1; ^13^C-NMR (125 MHz, methanol-*d*_4_): Table 2; and positive-ion HR-ESI-MS: *m/z* 777.1650 [M + Na]^+^ (calcd for C_40_H_34_O_15_Na: 777.1653).

### 3.7. Bioactivity Assays

The assays of evaluating nitric oxide (NO) production and cell viability on RAW264.7 cells were followed our studies before [9,10] and consulted literature [39]. RAW264.7 cells were seeded at a density of 5 × 10^4^ cells/well in 96-well plates for 12 h. Cells were treated with linderakoside F (**1**) in the presence of LPS (100 ng/mL) for 24 hours. Supernatants were collected and NO levels were determined using the Greiss reagent. Each of 100 μL of supernatant was mixed with 100 μL of Griess reagent (0.1% *N*-(1-naphthyl)-ethylenediamine dihydrochloride, 1% sulfanilamide, and 5% phosphoric acid) then incubated for 5 min at room temperature. The absorbance of the mixture was measured at 540 nm using a microplate reader (SpectraMax^®^ M2e, Molecular Devices, Sunnyvale, CA, USA). Culture media were used as blanks and the nitrite levels were determined by using a standard curve obtained from sodium nitrite (0–125 μM).

## 4. Conclusions

Despite being an endemic species of Lauraceae in Taiwan, there are not many reports yet on the phytochemistry and bioactivities of *L. akoensis*. Traditionally, *L. akoensis* is only used for ornamental purposes and some inflammation treatments. This study obtained three new flavonol acylglycosides, linderakosides F-H (compounds **1**–**3**) and three components **4**–**6** isolated from a natural product that first time, along with 25 known compounds, including two monoterpenoids **7**–**8**, 17 sesquiterpenoids **9**–**25**, and six steroids **26**–**31** which were isolated from this plant for the first time. Linderakoside F (**1**) displayed potential anti-inflammatory activity, with an IC_50_ value of 36.3 ± 3.2 μM. In this work, we discovered active components as potential lead compounds and additionally provided a scientific basis for the drug use of *L. akoensis*.

## Figures and Tables

**Figure 1 molecules-24-00563-f001:**
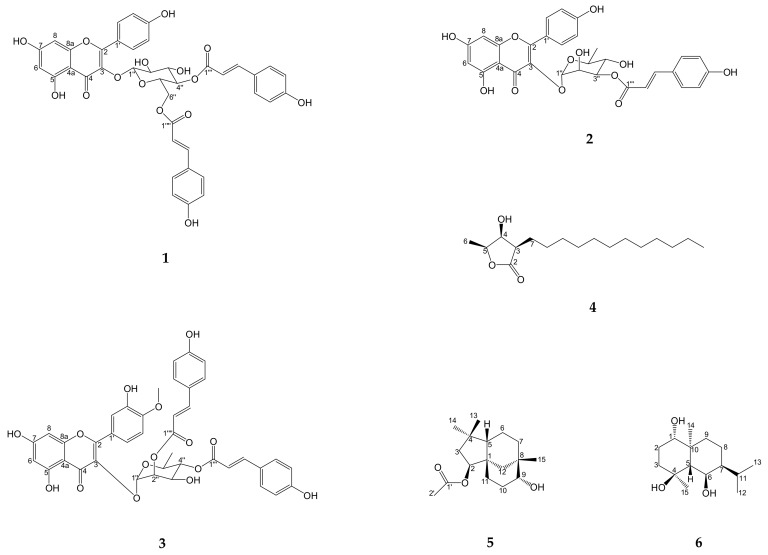
Compounds **1**–**6** isolated from the aerial part of *L. akoensis.*

**Figure 2 molecules-24-00563-f002:**
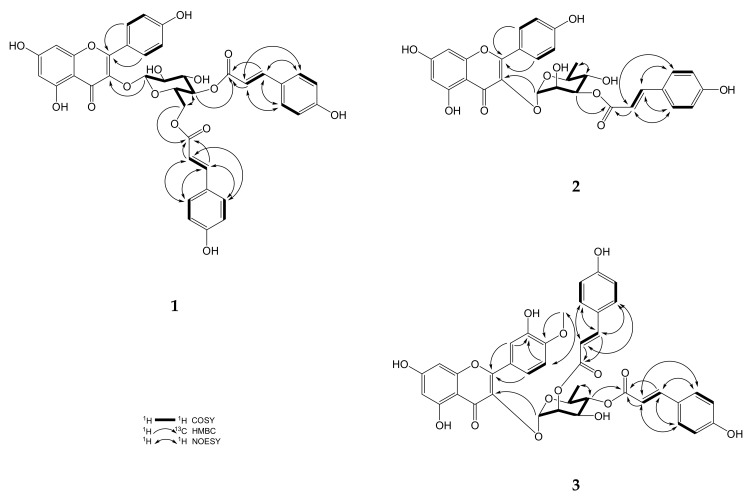
Selected key HMBC, COSY, and NOESY correlations of compounds **1**–**3**.

**Figure 3 molecules-24-00563-f003:**
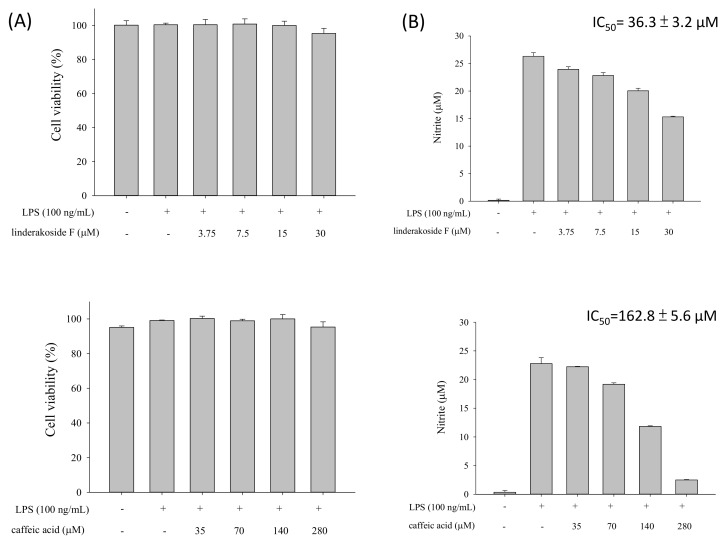
(**A**) Cytotoxicity of linderakoside F and caffeic acid in LPS-stimulated RAW264.7 cells. Cells were treated with linderakoside F at 3.75, 7.5, 15, 30 μM and caffeic acid at 35, 70, 140, 280 μM for 24 h, and cell viability was assayed by the MTT assay. Data were expressed as the means ± S.D. of three respectively experiments. (**B**) Effect of linderakoside F (**1**) and caffeic acid on NO production in LPS-stimulated RAW264.7 cells. Cells were incubated with LPS (100 ng/mL) in the presence of following doses at 3.75, 7.5, 15, 30 μM and 35, 70, 140, 280 μM of linderakoside F (**1**) and caffeic acid respectively for 24 h. Values were expressed as mean ± S.D. of three replicates. Mean with different letters represent significantly different (*p* < 0.05) by Scheffé’s method.

**Table 1 molecules-24-00563-t001:** ^1^H-NMR spectroscopic data of compounds **1**–**3** (in methanol-*d*_4_, 500 MHz) ^a^.

Position	Linderakoside F (1)	Linderakoside G (2)	Linderakoside H (3)
6	6.12 (1H, *br s*)	6.21 (1H, *br s*)	6.22 (1H, *d*, *J* = 2.0)
8	6.26 (1H, *br s*)	6.40 (1H, *br s*)	6.39 (1H, *d*, *J* = 2.0)
2′	7.97 (1H, *d*, *J* = 8.5)	7.84 (1H, *d*, *J* = 8.8)	7.40 (1H, *s*)
3′	6.82 (1H, *d*, *J* = 8.5)	6.97 (1H, *d*, *J* = 8.8)	-
5′	6.82 (1H, *d*, *J* = 8.5)	6.97 (1H, *d*, *J* = 8.8)	7.15 (1H, *d*, *J* = 8.9)
6′	7.97 (1H, *d*, *J* = 8.5)	7.84 (1H, *d*, *J* = 8.8)	7.41 (1H, *d*, *J* = 8.9)
1′′	5.36 (1H, *d*, *J* = 7.8)	5.47 (1H, *br s*)	5.71 (1H, *d*, *J* = 1.6)
2′′	3.73 (1H, *dd*, *J* = 9.4, 7.8)	4.44 (1H, *br s*)	5.55 (1H, *dd*, *J* = 3.4, 1.6)
3′′	3.59 (1H, *t*, *J* = 9.4)	5.13 (1H, *m*)	4.17 (1H, *dd*, *J* = 9.8, 1.6)
4′′	5.18 (1H, *t*, *J* = 9.4)	3.61 (1H, *t*, *J* = 9.3)	4.97 (1H, *t*, *J* = 9.8)
5′′	3.67 (1H, *m*)	3.44 (1H, *qd*, *J* = 5.5, 9.3)	3.31 (1H, *qd*, *J* = 6.2, 9.8)
6′′	4.25 (1H, *dd*, *J* = 12.5, 6.5) 4.34 (1H, *d*, *J* = 12.5)	0.97 (3H, *d*, *J* = 5.5)	0.85 (3H, *d*, *J* = 6.2)
2′′′	6.41 (1H, *d*, *J* = 15.9)	6.43 (1H, *d*, *J* = 15.7)	6.30 (1H, *d*, *J* = 16.0)
3′′′	7.68 (1H, *d*, *J* = 15.9)	7.72 (1H, *d*, *J* = 15.7)	7.60 (1H, *d*, *J* = 16.0)
5′′′	7.44 (1H, *d*, *J*= 8.5)	7.48 (1H, *d*, *J* = 8.7)	7.50 (1H, *d*, *J* = 8.6)
6′′′	6.81 (1H, *d*, *J* = 8.5)	6.81 (1H, *d*, *J* = 8.7)	6.82 (1H, *d*, *J* = 8.6)
8′′′	6.81 (1H, *d*, *J* = 8.5)	6.81 (1H, *d*, *J* = 8.7)	6.82 (1H, *d*, *J* = 8.6)
9′′′	7.44 (1H, *d*, *J* = 8.5)	7.48 (1H, *d*, *J* = 8.7)	7.50 (1H, *d*, *J* = 8.6)
2′′′′	6.06 (1H, *d*, *J* = 15.9)	-	6.42 (1H, *d*, *J* = 16.0)
3′′′′	7.39 (1H, *d*, *J* = 15.9)	-	7.70 (1H, *d*, *J*= 16.0)
5′′′′	7.26 (1H, *J* = 8.5)	-	7.50 (1H, *d*, *J* = 8.6)
6′′′′	6.80 (1H, *J* = 8.5)	-	6.85 (1H, *d*, *J* = 8.6)
8′′′′	6.80 (1H, *J* = 8.5)	-	6.85 (1H, *d*, *J* = 8.6)
9′′′′	7.26 (1H, *J* = 8.5)	-	7.50 (1H, *d*, *J* = 8.6)
OCH_3_	-	-	3.87 (3H, *s*)

^a^ The chemical shifts are expressed in δ ppm. The coupling constants (*J*) are expressed in Hz.

**Table 2 molecules-24-00563-t002:** ^13^C-NMR spectroscopic data of compounds **1**–**3** (in methanol-*d*_4_, 125 MHz).

Position	Linderakoside F (1)	Linderakoside G (2)	Linderakoside H (3)
2	159.3	159.6	159.4
3	135.4	136.2	134.1
4	179.3	176.5	179.6
4a	105.7	103.2	100.2
5	158.4	158.8	158.8
6	100.1	100.1	100.2
7	165.9	166.2	166.4
8	95.1	95.0	95.1
8a	162.9	163.4	163.4
1′	122.7	122.7	125.0
2′	132.4	132.1	131.5
3′	117.0	116.8	148.1
4′	161.2	161.8	152.0
5′	117.0	116.8	114.9
6′	132.4	132.1	131.5
1′′	104.1	103.2	100.2
2′′	74.2	70.1	73.1
3′′	70.3	75.3	68.6
4′′	78.9	70.7	74.9
5′′	75.8	72.4	70.0
6′′	64.4	17.9	18.0
1′′′	169.2	169.1	168.7
2′′′	115.5	116.0	114.8
3′′′	146.9	147.0	147.6
4′′′	127.4	127.5	127.3
5′′′	131.3	132.1	131.5
6′′′	116.9	116.0	117.0
7′′′	161.3	161.5	161.6
8′′′	116.9	116.0	117.0
9′′′	131.3	132.1	131.5
1′′′′	168.9	-	168.4
2′′′′	114.8	-	114.9
3′′′′	146.7	-	147.6
4′′′′	127.2	-	127.3
5′′′′	131.3	-	131.5
6′′′′	116.2	-	117.0
7′′′′	161.6	-	161.6
8′′′′	116.2	-	117.0
9′′′′	131.3	-	131.5
OCH_3_	-	-	56.7

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
