# Peer review of "New Anti-inflammatory Flavonol Glycosides from Lindera akoensis Hayata"

_molecules, 2019, doi:10.3390/molecules24030563_

Round 1

Reviewer 1 Report

Dear authors

The manuscript molecules-428283 by Yang et al., describes the isolation and structural elucidation of 31 compounds (3 new compounds, 3 isolated for the first time from natural source and 25 known compounds) from hydroethanolic extract of Lindera akoensis Hayata aerial parts. The authors also evaluate the anti-inflammatory activity of compound 1 (linderakoside F) which was much more active than the caffeic acid used as a positive reference.

The subject of this manuscript fits the aim and scope of Molecules but it is the continuation of the works already published in 2013 and 2017 by the same group (ref 9 and 12). Using the same experimental methodology, the same extract, and the same polarity fractions, they isolated new compounds, also from the class of flavonols glycosides, and isolated known compounds from terpenoids and steroids classes. Like in previous work, here they assess the anti-inflammatory activity of isolated compound.

Thus the novelty and impact of this manuscript decreases significantly.

But, in-depth phytochemical studies of pharmacologically relevant species and the structural elucidation of new compounds is always relevant knowledge that could be published (although fragmentation of a single work should be avoided).

However, the manuscript exhibits several weaknesses identified below and in the attached file (it is not an exhaustive list) that must be eradicated.

-          My main concern is for the identification of the isolated compounds.

The authors should carefully review the structural elucidation presented, clearly showing the spectroscopic evidence that allows us to conclude the unambiguous structure of each compound. In fact, for several compounds, the authors do not clarify several key points of the structure.

For example, for compound 1, I can not accept the indication of the existence of a 3-O-α-L-glucose unit in structure. The L-glucose is a very scarce natural compound in higher plants and the authors do not show any spectroscopic evidence of the presence in the molecule of this specific enantiomer.

Where are the chemical or spectroscopic evidence that is a unit of glucose? Or where are the chemical or spectroscopic evidence that is an alpha anomeric configuration?

To the compound 2, the author must to prove the anomeric configuration (α or β) of the rhamnose unit and kaempferol unit (as they prove the presence of (E)-p-coumaroyl nucleus) to show the existence of AFZELIN unit.

-          The authors must to clarify and expose much more clearly the spectroscopic evidence that allows to elucidate the structure of the compounds. Please see notes in the attached file.

-          Please provide the standard deviation associated to each IC50 value present in the manuscript.

-          Introduction: the authors don’t provide all the relevant references about previous phytochemical work on Lindera akoensis Hayata aerial parts. (2015, Planta Medica 81(16), DOI: 10.1055/s-0035-1565423)

-          The item “3.3 Extraction and Isolation” doesn’t give all necessary details (please see the notes on attached file)

-          Lines 177-189: This paragraph should be presented before the discussion of anti-inflammatory activity, so that the phytochemical study is all together.

-          Following the IUPAC rules, the letters D and L (identification of the sugar series) should be 2 points size lower than the text; the double bond configuration at coumaroyl nucleus should be indicated in parenthesis -(E)- Please check the entire manuscript.

-          Authors should carefully review the text, correcting spelling errors, words run together, unformatting, and even grammatical constructions … Enter a space between the indication of the reference and the word that precedes this reference. Please check the entire manuscript. See other examples in the attached file.

-          Please carefully review the final references list and format it according to Molecules author instructions (use the standard abbreviated journal name; in article title, the Latin binominal botanical name must be in italic, delete de issue number, …). Please see some notes in the attached file.

Author Response

Response to Reviewer’s Comments and Suggestions

We thank all reviewers for many constructive, considerate, and specific comments. We have adapted or addressed all of their comments and suggestions in the itemized way as follows.

Comment:

1. The authors should carefully review the structural elucidation presented, clearly showing the spectroscopic evidence that allows us to conclude the unambiguous structure of each compound. In fact, for several compounds, the authors do not clarify several key points of the structure.

For example, for compound 1, I can not accept the indication of the existence of a 3-O-α-L-glucose unit in structure. The L-glucose is a very scarce natural compound in higher plants and the authors do not show any spectroscopic evidence of the presence in the molecule of this specific enantiomer. 

Response:

Thank you for your kind reminding. Sorry, we made a mistake. We agree with you and have corrected the correct information “D-glucose” in the manuscript.

Comment:

2. Where are the chemical or spectroscopic evidence that is a unit of glucose? Or where are the chemical or spectroscopic evidence that is an alpha anomeric configuration? 

Response:

Thank you for your kind suggestion. We improved the characterization of coupling constant for the above structure. (line 75-77.)

Comment:

3. To the compound 2, the author must to prove the anomeric configuration (α or β) of the rhamnose unit and kaempferol unit (as they prove the presence of (E)-p-coumaroyl nucleus) to show the existence of AFZELIN unit.

Response:

Thank you for your kind suggestion. Thank you for your kind suggestion. We improved the characterization of coupling constant for the above structure. (line 96-97.)

Comment:

4. The authors must to clarify and expose much more clearly the spectroscopic evidence that allows to elucidate the structure of the compounds. Please see notes in the attached file.

Response:

Thank you for your detailed suggestions. We have revised the manuscript in accordance with the attached file you provided.

Comment:

5. Please provide the standard deviation associated to each IC50 value present in the manuscript.

Response:

Thank you for your suggestions. We added standard deviation to the manuscript.

Comment:

6. Introduction: the authors don’t provide all the relevant references about previous phytochemical work on Lindera akoensis Hayata aerial parts. (2015, Planta Medica 81(16), DOI: 10.1055/s-0035-1565423).

Response:

Thank you for your careful exploration and reminding. We cited the above-mentioned literature in the introduction. (line 40; references 13.)

Comment:

7. The item “3.3 Extraction and Isolation” doesn’t give all necessary details (please see the notes on attached file)

Response:

Thank you for reminding. In accordance with your suggestion, we have refined the information in section 3.3.

Comment:

8. Lines 177-189: This paragraph should be presented before the discussion of anti-inflammatory activity, so that the phytochemical study is all together.

Response:

Thank you for your valuable comments. We have adjusted the content of the article to make it more systematic.

Comment:

9. Following the IUPAC rules, the letters D and L (identification of the sugar series) should be 2 points size lower than the text; the double bond configuration at coumaroyl nucleus should be indicated in parenthesis -(E)- Please check the entire manuscript.

Response:

Thank you for your careful reminder of my mistakes. The above mistakes have been corrected in the article.

Comment:

10. Authors should carefully review the text, correcting spelling errors, words run together, unformatting, and even grammatical constructions … Enter a space between the indication of the reference and the word that precedes this reference. Please check the entire manuscript. See other examples in the attached file.

Response:

Thank you for your careful reminder of my mistakes. The above mistakes have been corrected in the article.

Comment:

11. Please carefully review the final references list and format it according to Molecules author instructions (use the standard abbreviated journal name; in article title, the Latin binominal botanical name must be in italic, delete de issue number, …). Please see some notes in the attached file.

Response:

Thank you for your careful reminder of my mistakes. The above mistakes have been corrected in the article.

Reviewer 2 Report

In the original paper: “New flavanol aglycones from Lindera akoensis Hayata”, the isolation, identification of new compounds and anti-inflammatory activity of linderakoside F are presented. In my opinion, the manuscript, although it takes an interesting topic, requires many changes and additions.

1. The Abstract should contain not only the results but also the purpose of the work, the background, as well as the briefly described research methodology.

2. The Introduction should be restructured in a way to show clear goals, targets of your study.

3.In the discussion chapter the part concerning biological assay, it's not developed enough. Please enter relevant literature, references and refer to the reference substance. 

4. In “Results and Discussion” Section there are the fragments that should be included in the Experimental Section.

5. The methodology of MTT and anti-inflammatory assays must be more detailed. 

6. The data describing the isolated compounds are scattered in Sections 2 and 3. I suggest putting them in one place.

Author Response

Response to Reviewer’s Comments and Suggestions

We thank all reviewers for many constructive, considerate, and specific comments. We have adapted or addressed all of their comments and suggestions in the itemized way as follows.

Comment:

1. The Abstract should contain not only the results but also the purpose of the work, the background, as well as the briefly described research methodology. 

Response:

Thank you for your suggestion. In the new version of the manuscript, we have re-structured the abstract.

Comment:

2. The Introduction should be restructured in a way to show clear goals, targets of your study. 

Response:

Thank you for your suggestion. In the new version of the manuscript, we refined the introduction.

Comment:

3. In the discussion chapter the part concerning biological assay, it's not developed enough. Please enter relevant literature, references and refer to the reference substance.

Response:

Thank you for your kind reminder of our shortcomings. We refined the “Results and Discussion” section according to your suggestion and literature in the new version manuscript. (line 175; references 36-38.)

Comment:

4. In “Results and Discussion” Section there are the fragments that should be included in the Experimental Section.

Response:

Thank you for your kind reminder. We re-structured and optimized the above sections in the new version manuscript.

Comment:

5. The methodology of MTT and anti-inflammatory assays must be more detailed.

Response:

Thank you for your suggestion. We improved section 3.7 in the new version manuscript.

Comment:

6. The data describing the isolated compounds are scattered in Sections 2 and 3. I suggest putting them in one place.

Response:

Thank you for your suggestion. We deleted repeated information in section 2 and moved the detailed data to section 3 as much as possible.

Reviewer 3 Report

The manuscript presents a complex phytochemical analysis on the aerial parts of Lindera akoensis, an endemic species from Taiwan.  There are a lot of chemical data concerning the structure of the 6 identified compounds, reported for the first time in this paper.

In order to improve the scientific value of the manuscript, I have some recommandations: - the language should to be improved, because there are a lot of speech and grammatical mistakes, especially in the introduction and conclusions parts

- the title is not in perfect agreement with the research  and, in my opinion, it is not complete: the three identified compounds are not flavanols, but flavonols, and they are not aglycons, they are glycosides. Anyway, only three of the 6 identified compounds are flavonoid-derivatives, the other three have different structures. The evaluation of the anti-inflammatory activity is not mentioned in the title

-I think further studies are necessary to confirm the anti-inflammatory activity for compound 1 and to evaluate the activity of the other compounds, or even the activity of the aerial parts extract

- I wonder why the vegetable product was harvested in 2008 and the analysis were done after 10 years? The chemical composition of a natural product could be influenced by the preservation and the study become no relevant. We don't know if the same compounds are present in the natural product harvested last year!

Author Response

Response to Reviewer’s Comments and Suggestions

We thank all reviewers for many constructive, considerate, and specific comments. We have adapted or addressed all of their comments and suggestions in the itemized way as follows.

Comment:

1. The title is not in perfect agreement with the research  and, in my opinion, it is not complete: the three identified compounds are not flavanols, but flavonols, and they are not aglycons, they are glycosides. Anyway, only three of the 6 identified compounds are flavonoid-derivatives, the other three have different structures. The evaluation of the anti-inflammatory activity is not mentioned in the title. 

Response:

Thank you for your suggestion. We'll optimize the title to "New Anti-inflammatory Flavonol Glycosides from Lindera akoensis Hayata".

Comment:

2. I think further studies are necessary to confirm the anti-inflammatory activity for compound 1 and to evaluate the activity of the other compounds, or even the activity of the aerial parts extract. 

Response:

Thank you for reminding us of our shortcomings. For evaluating the anti-inflammatory mechanism of compound 1, we are still in the process of establishing a suitable model. The other compounds weigh too little to evaluate their activity or have no anti-inflammatory activity. We are seeking to work with other teams to evaluate their other biological activities. I am sorry that all the crude extract of L. akoensis was used for separation and purification, so its activity was not evaluated at that time.

Comment:

3. I wonder why the vegetable product was harvested in 2008 and the analysis were done after 10 years? The chemical composition of a natural product could be influenced by the preservation and the study become no relevant. We don't know if the same compounds are present in the natural product harvested last year!

Response:

Thank you for your question. L. akoensis collected in 2008, extraction, separation, purification and structure identification were completed before 2012, bioassays completed in 2013. From 2014 to June 2018, I worked in a private enterprise, but did not serve in an educational institution. Since September 2018, I have worked in Longyan university to complete the consolidation and publication of data.

Round 2

Reviewer 1 Report

Dear authors

The manuscript molecules-428283 by Yang et al. was improved significantly and the main reviewer suggestions were taken into account.

However, the manuscript exhibits yet some minor points to be corrected:

-          Lines 71-72: The grammatical construction of the following sentence is incorrect. “The characteristic of kaempferol observed on 1H-NMR consistent with literature [14], the…”

Where is the main verb of this sentence? It should be “The characteristic signals of kaempferol observed on 1H-NMR are consistent with literature [14], the…”

-          Lines 75: it should be “The axial-axial coupling constant” instead “The axil-axil coupling constant”

-          Line 178: it should be “…control. Linderakoside F (1) showed in vitro anti-inflammatory activity since it decreases the LPS-stimulated…” instead “…control. linderakoside F (1) showed in vitro anti-inflammatory activity decrease the LPS-stimulated…”

-           The text continues to present some misformatted symbols (see lines 95, 240)

-          Line 262: Where is “Traditionally, L. akoensis only used for ornamentation…” it should be “Traditionally, L. akoensis is only used for ornamentation…”

Author Response

Dear reviewer,

Enclosed please find the revised version of our manuscript, entitled New Anti-inflammatory Flavonol Glycosides from Lindera akoensis Hayata (molecules-428283) by Chung-Ping Yang, Pei-Hsin Shie, Guan-Jhong Huang, Shih-Chang Chien and Yueh-Hsiung Kuo, which has been recently reviewed by your editorial board. We appreciate and agree much of the positive and constructive comments and suggestions offered by you. We also have addressed in the revised manuscript and here the specific questions, critics, and suggestions. With these efforts, we believe that we have clearly upgraded and improved the quality of our manuscript, and effectively addressed the reviewer’s comments. We hope that you would agree that the manuscript now merits publication in Molecules.

Sincerely yours,

Chung-Ping Yang, Associate Prof.

College of Life Sciences, Longyan University, No.1 North Dongxiao Road, Longyan Fujian, 364012, P. R. China.

Tel: +86-597-2797255

Mobile: +86-13225908986 (China); +886-912668570 (Taiwan)

E-mail: [email protected]; [email protected]

Comment
:

1. Lines 71-72: The grammatical construction of the following sentence is incorrect. “The characteristic of kaempferol observed on 1H-NMR consistent with literature [14], the…”

Where is the main verb of this sentence? It should be “The characteristic signals of kaempferol observed on 1H-NMR are consistent with literature [14], the…”

Response:

Thank you for discovering my mistake carefully. I have corrected the above error in the new version of the manuscript. (Lines 71-72)

Comment:

2. Lines 75: it should be “The axial-axial coupling constant” instead “The axil-axil coupling constant”

Response:

Thank you for discovering my mistake carefully. I have corrected the above error in the new version of the manuscript. (Lines 75)

Comment:

3. Line 178: it should be “…control. Linderakoside F (1) showed in vitro anti-inflammatory activity since it decreases the LPS-stimulated…” instead “…control. linderakoside F (1) showed in vitro anti-inflammatory activity decrease the LPS-stimulated…”

Response:

Thank you for discovering my mistake carefully. I have corrected the above error in the new version of the manuscript. (Lines 178)

Comment:

4. The text continues to present some misformatted symbols (see lines 95, 240)

Response:

Thank you for discovering my mistake carefully. I have corrected the above error in the new version of the manuscript. (Lines 95, 240, 245)

Comment:

5. Line 262: Where is “Traditionally, L. akoensis only used for ornamentation…” it should be “Traditionally, L. akoensis is only used for ornamentation…”

Response:

Thank you for discovering my mistake carefully. I have corrected the above error in the new version of the manuscript. (Lines 262)

Reviewer 2 Report

As the authors addressed the reviewers comments, I suggest acceptance of the manuscript.

Author Response

Dear reviewer,
Thank you for your corrections and suggestions, which will be very helpful for our future research and submission.

Warm Regards,

Chung-Ping Yang, Associate Prof.

College of Life Sciences, Longyan University, No.1 North Dongxiao Road, Longyan Fujian, 364012, P. R. China.

Tel: +86-597-2797255

Mobile: +86-13225908986 (China); +886-912668570 (Taiwan)

E-mail: [email protected]; [email protected]